# Monosynaptic tracing maps brain-wide afferent oligodendrocyte precursor cell connectivity

Christopher W Mount[1,2,3†], Belgin Yalçın[1†], Kennedy Cunliffe-Koehler[1], Shree Sundaresh[1], Michelle Monje[1,4,5,6*]

[1]Department of Neurology, Stanford University, Stanford, United States; [2]Medical Scientist Training Program, Stanford University, Stanford, United States; [3]Neurosciences Graduate Program, Stanford University, Stanford, United States; [4]Institute for Stem Cell Biology and Regenerative Medicine, Stanford University, Stanford, United States; [5]Department of Pathology, Stanford University, Stanford, United States; [6]Department of Pediatrics, Stanford University, Stanford, United States

**Abstract** Neurons form bona fide synapses with oligodendrocyte precursor cells (OPCs), but the circuit context of these neuron to OPC synapses remains incompletely understood. Using monosynaptically-restricted rabies virus tracing of OPC afferents, we identified extensive afferent synaptic inputs to OPCs residing in secondary motor cortex, corpus callosum, and primary somatosensory cortex of adult mice. These inputs primarily arise from functionally-interconnecting cortical areas and thalamic nuclei, illustrating that OPCs have strikingly comprehensive synaptic access to brain-wide projection networks. Quantification of these inputs revealed excitatory and inhibitory components that are consistent in number across brain regions and stable in barrel cortex despite whisker trimming-induced sensory deprivation.
DOI: https://doi.org/10.7554/eLife.49291.001

*For correspondence:
mmonje@stanford.edu

†These authors contributed equally to this work

Competing interests: The authors declare that no competing interests exist.

## Introduction

Excitatory and inhibitory synapses between neurons and OPCs are well-established and the ultra-structural and electrophysiological features of these 'axon->glial' synapses have been investigated in slice preparations, generally by evoking potentials in local fiber bundles (*Ziskin et al., 2007*; *De Biase et al., 2010*; *Kukley et al., 2007*; *Lundgaard et al., 2013*). However, the afferent projections from neurons to OPCs providing this synaptic input have not been systematically mapped, and thus our understanding of the neuronal territories accessed by neuron->OPC synapses has been limited. Recent evidence has demonstrated that neuronal activity robustly regulates OPC proliferation, oligodendrogenesis, and myelination in both juvenile and adult rodents (*Gibson et al., 2014*; *Mitew et al., 2018*; *Hughes et al., 2018*) and also influences axon selection during developmental myelination in zebrafish (*Mensch et al., 2015*; *Hines et al., 2015*). These activity-regulated responses of oligodendroglial cells have been shown to confer adaptive changes in motor function (*Gibson et al., 2014*), are necessary for some forms of motor learning (*McKenzie et al., 2014*; *Xiao et al., 2016*) and contribute to cognitive behavioral functions such as attention and short-term memory (*Geraghty et al., 2019*). Appreciation for this plasticity of myelin has stoked interest in the axon->glial synapse as a means by which OPCs could detect and integrate activity-dependent neuronal signals. Here, we employ a modified rabies virus-based monosynaptically-restricted trans-synaptic retrograde tracing strategy to elucidate a map of neuronal synaptic inputs to OPCs in the corpus callosum (CC), secondary motor cortex (MOs), and primary somatosensory cortex (SSp) in

vivo. We find brain-wide, functionally-interconnected inputs to OPCs and that the degree of this connectivity is stable across brain regions and is maintained despite whisker trimming-induced sensory deprivation in barrel cortex at the timepoint examined.

## Results

### Development and validation of retrograde monosynaptic OPC tracing strategy

Owing to the lack of viral tools to achieve specific transgene expression in OPCs, we employed a transgenic strategy by crossing *Pdgfra::*CreER mice (*Kang et al., 2010*), which permit OPC-specific Cre recombinase expression, with a Cre-inducible RABVgp4/TVA mouse (*Takatoh et al., 2013*). Offspring of this cross express rabies virus glycoprotein 4 (gp4) and the avian TVA receptor specifically in OPCs upon tamoxifen administration (*Pdgfra*::CreER-(gp4-TVA)^fl). Subsequently, we performed stereotaxic injection of ASLV-A (EnvA)-pseudotyped gp4-deleted rabies virus encoding EGFP (SADΔG-EGFP(EnvA)). EnvA's highly specific interaction with the TVA receptor ensures restriction of primarily-transduced cells – hereafter named starter cells – to the *Pdgfra+* OPC cell population. Because these OPCs also express rabies gp4, virions can be assembled within these starter cells and spread retrogradely across single synaptic connections to presynaptic input neurons; however, because these input neurons do not express gp4, there is no additional spread of virus beyond these monosynaptic connections (*Figure 1A*) (*Wickersham et al., 2007*). A caveat to this approach is that OPCs that differentiate to oligodendrocytes (*Ye et al., 2009*) between tamoxifen administration and virus injection would still be susceptible to infection; likewise, infected OPCs that undergo differentiation could skew histological assessment of input to starter cell ratios. To mitigate these concerns, we followed a narrow injection time course (*Figure 1A*) beginning in adult (6 month old) mice, when rates of OPC differentiation are substantially lower than in juveniles (*Young et al., 2013*).

Viral injection occured three days following a single dose of tamoxifen. At 5 days following injection of SADΔG-EGFP(EnvA) into the genu of the corpus callosum inferior to the cingulum bundle, we observed substantial labeling of presynaptic neuronal inputs (*Figure 1B*). In control (gp4-TVA)^fl mice lacking the *Pdgfra*::CreER transgene, injection of tamoxifen and modified rabies virus achieved only minor, local background labeling expected to result from small fractions of EnvA negative viral particles (*Figure 1C*). Pdgfra+/Olig2+/EGFP+ starter cells were present in the injection site (*Figure 1D*), while other glial subtypes including Gfap+ white matter astrocytes as well as Iba1+ macrophages and microglia were EGFP negative, confirming the specificity of glial infection to the targeted OPC population (*Figure 1—figure supplement 1A,B*). Immunostaining for Cre expression at this timepoint confirmed previously-reported driver specificity to OPCs (*Kang et al., 2010*), with no expression of Cre in NeuN+ neurons in this context (*Figure 1E*). Thus, the starter cell population is limited to the oligodendroglial lineage, with no evidence of non-synaptic 'leak' of virus into other cell populations. While viral infection did result in limited toxicity to infected OPCs as suggested by morphology, expression of the identifying markers, Pdgfra and Olig2, was retained (*Figure 1—figure supplement 1C*). To further confirm specificity of OPC transduction, we performed comparison injections with SADΔG-mCherry lacking EnvA – and therefore capable of unrestricted transduction – into *Pdgfra*::CreER and Cre WT control animals (*Figure 1—figure supplement 1D*). In stark contrast to (EnvA)-bearing virus, broad cortical transduction is observed in both animals. This difference in transduction patterns between EnvA-bearing virus and non-psuedotyped virus indicates that TVA expression in OPCs successfully restricts EnvA-bearing virus transduction.

### OPCs in corpus callosum receive brain-wide synaptic input

Quantification of input neurons revealed extensive neuronal territories that synapse onto starter OPCs in the corpus callosum (*Figure 2A,B*). Summing all inputs identified to these white matter OPCs, viral input/starter ratios in this context are approximately 23 (slope of linear regression 23.46 ± 2.8 standard error, *Figure 2C*), with neuronal input cells clearly identifiable by morphology and EGFP expression (*Figure 2D*). To corroborate this ratio, we performed immunofluorescence staining of PSD95 puncta in the corpus callosum and quantified colocalized puncta with Pdgfra+ OPCs with the aid of 3D modeling software (*Figure 2—figure supplement 1A*). Assessing 95 cells across nine animals, we found a distribution of PSD95-colocalized puncta from 0 to 67, with a mean

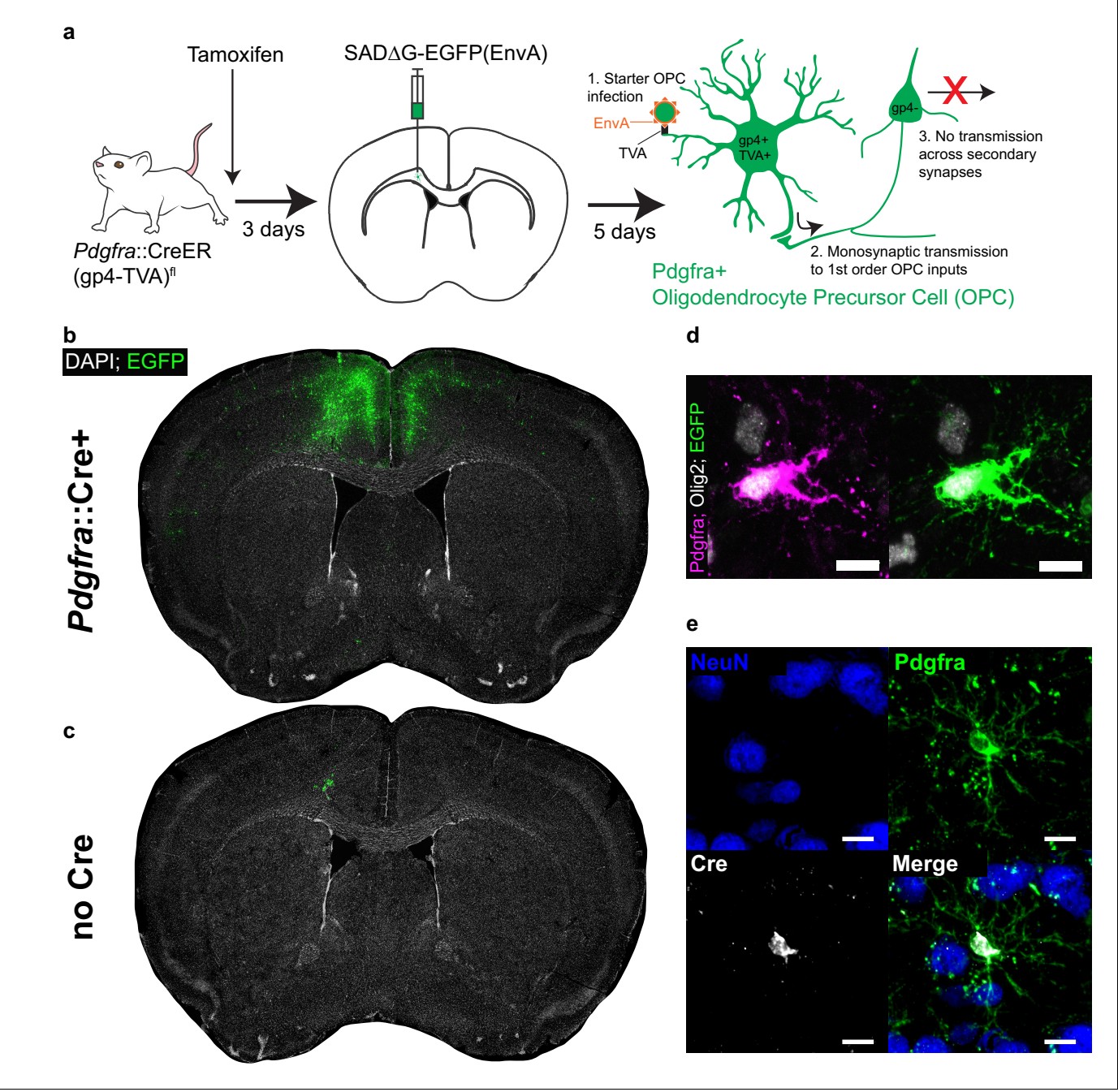

**Figure 1.** Monosynaptically-restricted rabies virus enables tracing of synaptic inputs to OPCs. (**a**) Outline of experimental strategy used to label inputs to Pdgfra+ OPCs. (**b**) Injection of SADΔG-EGFP(EnvA) into sub-cingular corpus callosum results in widespread labeling of EGFP+ input neurons (representative injection site image from n = 10 animals. Green = EGFP, white = DAPI). (**c**) Injection of SADΔG-EGFP(EnvA) into animals lacking *Pdgfra*::CreER driver allele results in only minimal transduction, likely resulting from minimal quantities of EnvA- viral particles (representative image of n = 4 animals. Green = EGFP, white = DAPI). (**d**) Pdgfra+/Olig2+ OPC starter cells (left) are transduced with SADΔG-EGFP(EnvA) (right, same cell. Magenta = Pdgfra, white = Olig2, green = EGFP). (**e**) Immunostaining confirms Cre recombinase expression in Pdgfra+ OPCs (green) but not NeuN+ neurons (blue). Scale bars in (**d,e**) represent 10 microns.

DOI: https://doi.org/10.7554/eLife.49291.002

The following figure supplement is available for figure 1:

**Figure supplement 1.** SADΔG-EGFP(EnvA) specificity and starter characterization.
DOI: https://doi.org/10.7554/eLife.49291.003

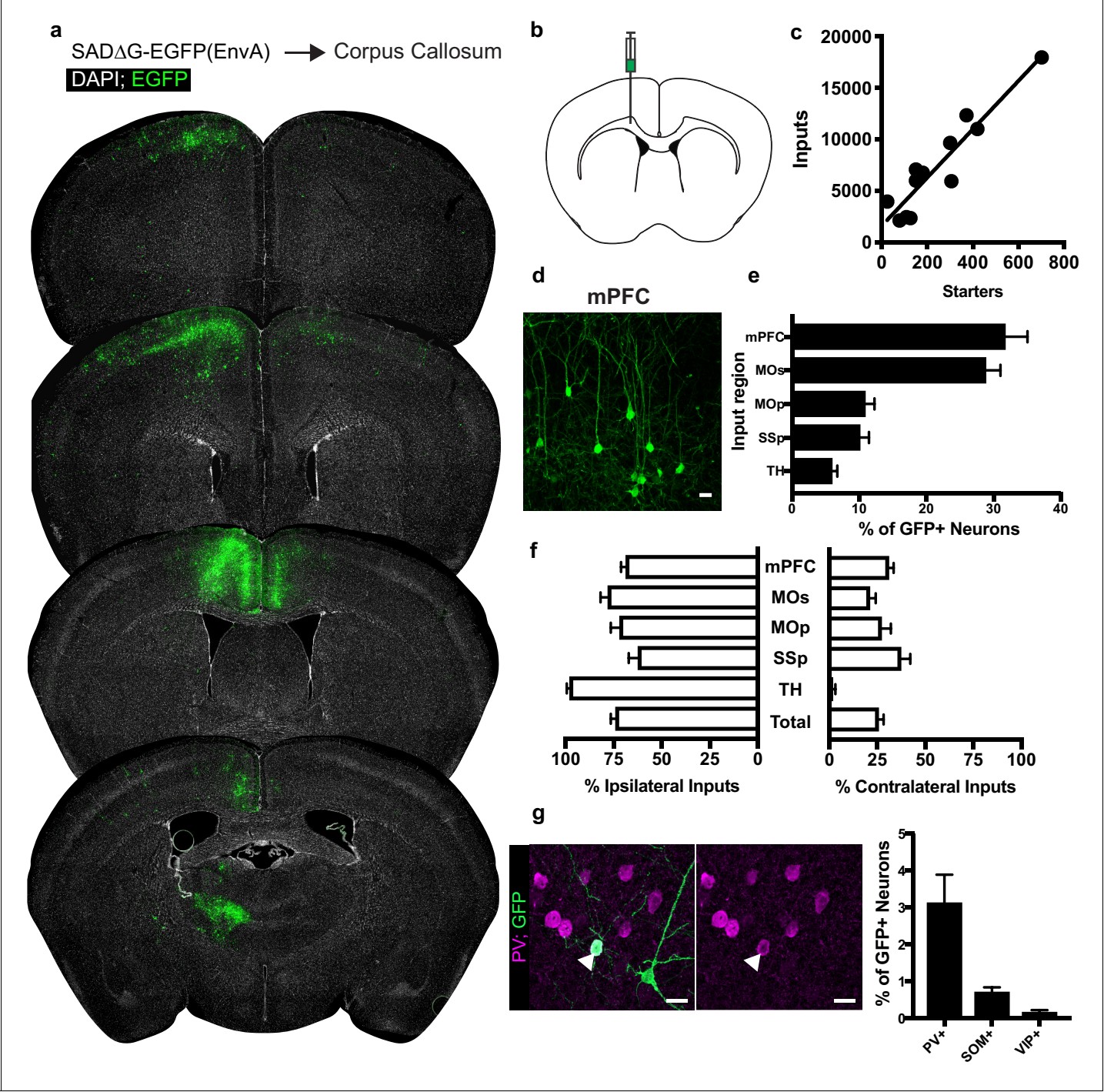

**Figure 2.** Neuronal inputs to callosal OPCs arise from functionally interconnected cortical and thalamic areas. (**a**) Representative sections of neuronal input labeling to OPCs following stereotaxic injection of SADΔG-EGFP(EnvA) to corpus callosum underlying the secondary motor area. Green = EGFP, white = DAPI. (**b**) Schematic of injection site. (**c**) Linear regression fit of neuronal input/Pdgfra+ OPC starter cells. Each point represents one animal, n = 12 animals, $R^2$ = 0.8732, slope = 23.46 ± 2.8 standard error. (**d**) Representative confocal micrograph of EGFP+ (green) input neurons in medial prefrontal (mPFC) cortex. (**e**) Inputs to callosal OPCs largely arise from frontal association cortices but also include primary motor and somatosensory areas and thalamic nuclei. Each bar represents mean input percentage, error bars indicate SEM, n = 10 total. mPFC = medial prefrontal cortex (anterior cingulate, prelimbic, infralimbic regions), MOs = secondary motor area, MOp = primary motor area, SSp = primary somatosensory area, TH = thalamus (including all thalamic nuclei). (**f**) Percent of input neurons ipsilateral or contralateral to the OPC starter cells. Bars indicate mean, error bars indicate SEM, n = 10 animals. (**g**) Representative image of parvalbumin+ (PV+, magenta) GFP+ (green) input neuron (arrowhead) and quantification of

*Figure 2 continued on next page*

*Figure 2 continued*

percentage of input neurons that co-label with immunofluorescence makers for PV, somatostatin (SOM), or vasoactive intestinal peptide (VIP). Bars represent mean, error bars indicate SEM, n = 6 animals total. Scale bars in (**d,g**) represent 20 microns.

DOI: https://doi.org/10.7554/eLife.49291.004

The following figure supplement is available for figure 2:

**Figure supplement 1.** PSD95 puncta colocalization with OPCs.

DOI: https://doi.org/10.7554/eLife.49291.005

of 21.62 puncta ±2.8 standard error per OPC (*Figure 2—figure supplement 1B,C*). Both myelinated and non-myelinated input axons are present as indicated by immunofluorescence staining for para-nodal CASPR (*Figure 1—figure supplement 1E*). Measuring starter OPC depth relative to the pia confirmed their subcortical location (*Figure 1—figure supplement 1F*). Neuronal inputs to OPCs in the genu of the corpus callosum inferior to the cingulum bundle are concentrated in dorsal and ventral mPFC (defined here to include anterior cingulate, pre- and infralimbic regions) and secondary motor cortex (MOs) (*Figure 2E*). Inputs from primary motor (MOp) and primary somatosensory (SSp) cortices are also present, along with substantial connectivity from the thalamus (TH, *Figure 2E*). These thalamic inputs are most consistently localized to ventroanterolateral (VAL), anteromedial (AM), and anterodorsal (AD) nuclei, consistent with thalamocortical projection neurons targeting motor and prefrontal cortical areas (Figure 5). The majority of inputs identified arise ipsilateral to the viral injection site; however, the relatively high contribution of inputs arising in the contralateral mPFC combined with high overall labeling densities in this region suggest that callosal OPCs are substantially innervated by contralateral intracortical projections (*Figure 2F*). By contrast, thalamic inputs are ipsilaterally restricted, further supporting the monosynaptic restriction of viral labeling (*Figure 2F*).

While GABAergic inputs to OPCs have been described (*Kukley et al., 2008*; *Lin and Bergles, 2004*), the majority of evidence for neuron-OPC synaptic connectivity in the corpus callosum arises from recordings of glutamatergic excitation either spontaneously or following callosal fiber stimulus (*Ziskin et al., 2007*; *De Biase et al., 2010*). Immunostaining for characteristic non-overlapping cortical inhibitory subpopulation markers accounting for the majority of total cortical inhibitory neurons (*Pfeffer et al., 2013*) – parvalbumin (PV), vasoactive intestinal peptide (VIP), and somatostatin (SOM) – revealed PV+/GFP+ co-labeled inputs encompassing approximately 3% of inputs to OPCs in CC (*Figure 2G*). The majority of these PV+GFP+ inputs were present ipsilaterally in the overlying MOs/mPFC. SOM or VIP co-labeled GFP+ input neurons comprised 1% or less of total inputs to OPCs in the CC. The excitatory to inhibitory neuron ratio of inputs to callosal OPCs is ~20:1, with inhibitory neurons defined by PV, VIP or SOM-expression.

## OPCs in premotor cortex receive synaptic input from premotor cortical and thalamic neurons

Examining the afferents to cortical OPCs in the secondary motor (premotor, MOs, M2) cortex, injection of SADΔG-EGFP(EnvA) into MOs of *Pdgfra*::CreER-(gp4-TVA)[fl] mice (*Figure 3B*) again resulted in infection of Pdgfra+/EGFP+ starter OPCs. Labeled input neurons were strikingly predominant within cortical territory defining the boundaries of MOs (*Figure 3A,D*). Input to starter cell ratios for these gray matter OPCs were not substantially different from those in CC-injected animals (slope of best-fit linear regression = 18.76 ± 4.4 standard error, *Figure 3C*). Beyond MOs, a smaller fraction of inputs arise from primary motor cortex (MOp), nearby medial prefrontal cortex (mPFC), and to a lesser extent, projections from SSp, and thalamocortical projection neurons (*Figure 3D,E*), illustrating brain-wide premotor circuit inputs to premotor cortical OPCs. Immunostaining for markers of interneuron identity revealed PV+/GFP+ inputs averaging 6% of total input neurons to mPFC OPCs, while SOM+/GFP+ or VIP+/GFP+ costaining was present in approximately 1% or less of total inputs (*Figure 3F*). Input neurons to MOs OPCs are primarily ipsilateral, with a smaller proportion of afferent projections arising contralaterally than observed in OPCs within the CC (*Figure 3G*). Like CC OPCs, the excitatory to inhibitory ratio of inputs to premotor cortical OPCs is ~20:1, with inhibitory neurons defined by PV, VIP or SOM-expression.

## OPCs in primary somatosensory cortex receive synaptic input from ipsilateral sensory cortex and thalamic neurons

To assess whether the pattern of cortical OPC inputs arising from local cortical neurons and functionally-associated thalamic nuclei was specific to MOs, we injected SADΔG-EGFP(EnvA) into primary somatosensory cortex (SSp) of *Pdgfra*::CreER-(gp4-TVA)[fl] mice (*Figure 4B*). As in MOs and CC, this resulted in primary infection of Pdgfra+/EGFP+ starter OPCs, and as in MOs, inputs were confined primarily to sensory cortical territory (SSp) (*Figure 4A*). Input to starter cell ratios at this site did not differ significantly from injections in CC or MOs (slope of best-fit linear regression = 22.57 ± 3.8 standard error, *Figure 4C*). Examination of GFP+ input neurons revealed inputs arising primarily from SSp across multiple cortical layers and from the thalamus (*Figure 4D,E*). In contrast to OPCs residing in CC, and to a lesser extent MOs, input neurons to OPCs in SSp are almost entirely ipsilaterally-restricted, and there is a small (<5%) contribution of input neurons from mPFC, MOs, or MOp. As in MOs, immunostaining for markers of interneuron identity revealed approximately 4% of GFP+ input neurons colabeled with PV, while SOM+ or VIP+ inputs comprised 1% or less of total GFP+ inputs (*Figure 4F*). Like CC and promotor cortex OPCs, the excitatory to inhibitory ratio of inputs to somatosensory cortical OPCs is ~20:1, with inhibitory neurons defined by PV, VIP or SOM-expression.

## Thalamic input neurons to OPCs arise from functionally-related thalamic nuclei

For OPCs in all brain regions studied, a substantial fraction of synaptic inputs arise from thalamic neurons. To assess whether these thalamic inputs arise from functionally-related nuclei, we registered acquired image tiles to the Allen Institute reference adult mouse brain atlas (*Lein et al., 2007*) and localized identified GFP+ inputs (*Figure 5*). Thalamic projections providing synaptic input to OPCs located in the corpus callosum underlying primary and secondary motor cortex arise primarily from ventral anterior-lateral (VAL) and anteromedial (AM) nuclei, consistent with known projections to motor planning territories (*Figure 5A*), along with projections from the anterodorsal (AD) nucleus. Strikingly, thalamic inputs to MOs OPCs also arose primarily from VAL and AM nuclei (*Figure 5B*). This is largely distinct from thalamic projections to SSp OPCs, which arise primarily within ventral posterolateral (VPL) and ventral posteromedial (VPM) regions, consistent with known projections to somatosensory targets (*Figure 5C*). Together, this suggests that particularly in the case of cortical OPCs, these thalamocortical synaptic inputs arise from functionally-related thalamic nuclei, consistent with expected thalamocortical projection patterns.

## Total synaptic connectivity to OPCs is consistent across brain regions despite reduced input neuron activity

To assess whether the degree of synaptic connections to OPCs varied across the injection sites assessed, we compared the average neuronal input ratios, assessed as the slope of the best-fit linear regression to total GFP+ inputs versus starter Pdgfra+/GFP+ OPCs. We found no significant difference in the synaptic input ratios between OPCs in the CC, MOs, or SSp (*Figure 6A*). To assess whether perturbations of synaptic input activity might modify the degree of synaptic connectivity, we performed daily whisker trimming of *Pdgfra*::CreER-(gp4-TVA)[fl] mice for 11 days prior to tamoxifen injection (*Figure 6B*). Anticipating that input activity to the cortical barrel field would be reduced in whisker-trimmed animals, we then injected SADΔG-EGFP(EnvA) into barrel field of trimmed and matched untrimmed control animals. Five days after viral injection, the animals were euthanized and total GFP+ input neurons and Pdgfra+/GFP+ starter OPCs were quantified. We found no significant difference in neuronal input to starter ratio between whisker-trimmed and untrimmed control animals as assessed by the slope of best-fit linear regression (*Figure 6C,D*). Moreover, we found no significant difference in the distribution of input neurons between trimmed and untrimmed animals, with primarily somatosensory cortical inputs and approximately 10% of inputs arising from thalamus (*Figure 6E*). Quantification of immunostaining for interneuron markers revealed PV+GFP+ inputs in equal proportion (3–4%) in trimmed and untrimmed groups (*Figure 6F*), indicating an unchanged excitatory to inhibitory (PV+ neuron) ratio of OPC inputs regardless of whisker trimming at this time point. Taken together, neither OPC location across white and gray matter territories, nor

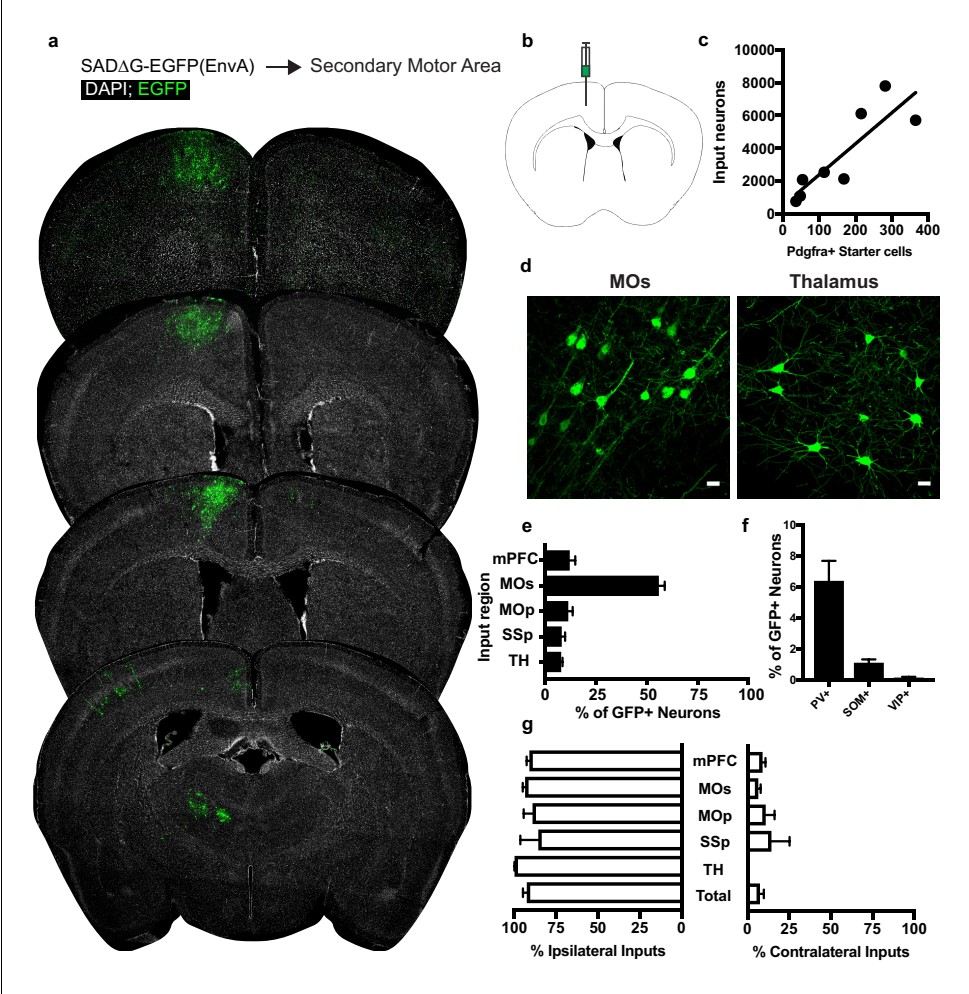

**Figure 3.** Circuit-specific cortical and thalamic neuronal inputs to OPCs in secondary motor area (MOs). (a) Representative sections of neuronal input labeling to OPCs following stereotaxic injection of SADΔG-EGFP(EnvA) to MOs. Green = EGFP, white = DAPI. (b) Schematic of injection site. (c) Linear regression fit of neuronal input/Pdgfra+ starter cells. Each point represents one animal, $R^2$ = 0.7486, slope = 18.76 ± 4.4 standard error). (d) Representative confocal micrographs of EGFP+ (green) input neurons in secondary motor cortex (MOs) and thalamus. (e) Inputs to gray matter OPCs found in MOs are chiefly located within MOs, n = 8 animals total. mPFC = medial prefrontal cortex (anterior cingulate, prelimbic, infralimbic regions), MOs = secondary motor area, MOp = primary motor area, SSp = primary somatosensory area, TH = thalamus (including all thalamic nuclei). (f) Percentage of input neurons that co-label with immunofluorescence makers for parvalbumin (PV), somatostatin (SOM), or vasoactive intestinal peptide (VIP), n = 5 animals total. (g) Percent of input neurons ipsilateral or contralateral to OPC starter cells. Bars indicate mean, error bars indicate SEM, each point represents an individual animal (n = 8). Scale bars in (d) represent 20 microns.
DOI: https://doi.org/10.7554/eLife.49291.006

modification of input activity in barrel field by whisker trimming modified the quantity or pattern of neurons providing synaptic input to OPCs at this time point.

## Discussion

Substantial progress in characterizing the electrophysiological properties of neuron->OPC synapses has yet to clarify their potential role in modulating oligodendrocyte lineage dynamics and ultimately animal behavior. In particular, prior to this work little was known regarding the extent of neuronal input territories to OPCs beyond local neurons and fiber bundles accessible in a slice preparation. Using a monosynaptically restricted trans-synaptic retrograde tracing system, we have now elucidated a map of neuronal input territories to OPCs in three distinct regions of the mouse brain. OPCs in these territories – selected due to previously reported changes in local oligodendrocyte lineage

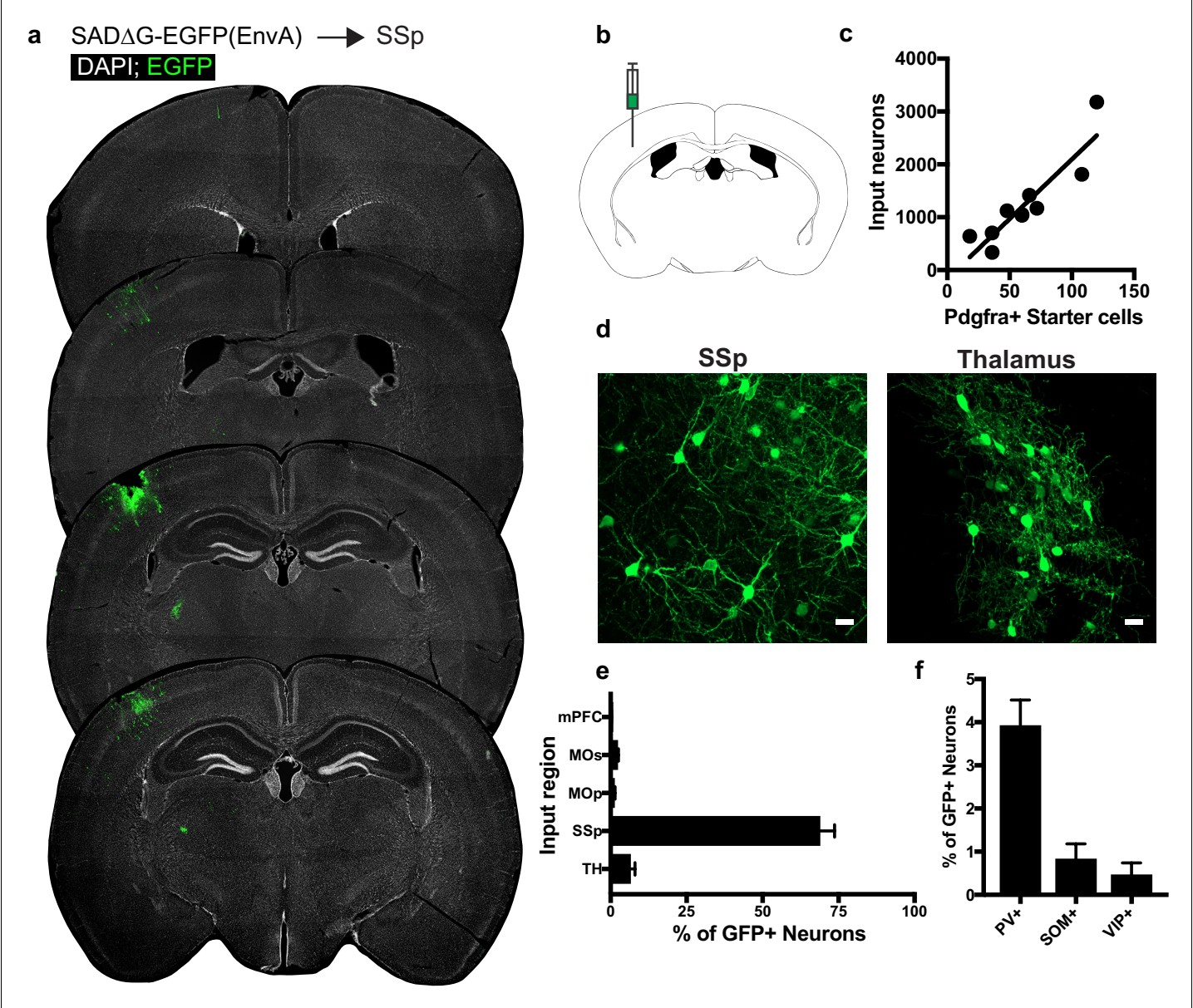

**Figure 4.** Circuit-specific cortical and thalamic neuronal neuronal inputs to SSp. (a) Representative sections of neuronal input labeling to OPCs following stereotaxic injection of SADΔG-EGFP(EnvA) to SSp. Green = EGFP, white = DAPI. (b) Schematic of injection site. (c) Linear regression fit of neuronal input/Pdgfra+ starter cells. Each point represents one animal, $R^2$ = 0.8145, slope = 22.57 ± 3.8 standard error). (d) Representative confocal micrographs of EGFP+ (green) input neurons in primary somatosensory cortex (SSp) and thalamus. (e) Inputs to gray matter OPCs found in SSp are chiefly located within SSp. n = 9 animals total. mPFC = medial prefrontal cortex (anterior cingulate, prelimbic, infralimbic regions), MOs = secondary motor area, MOp = primary motor area, SSp = primary somatosensory area, TH = thalamus (including all thalamic nuclei). (f) Percentage of input neurons that co-label with immunofluorescence makers for parvalbumin (PV), somatostatin (SOM), or vasoactive intestinal peptide (VIP), n = 5 animals. Bars indicate mean, error bars indicate SEM. Scale bars in (d) represent 20 microns.

DOI: https://doi.org/10.7554/eLife.49291.007

dynamics in response to neuronal activity – all receive brain-wide synaptic input. Strikingly, the ratio of input neurons to starter OPCs is consistent across MOs, SSp, and CC, despite the greater local axon density in CC and despite higher OPC turnover rates in CC than either cortical territory. This suggests that regulation of the number of neuron->OPC synapses may be intrinsic to the OPC rather than specified by local neurons or other microenvironmental factors in these brain regions.

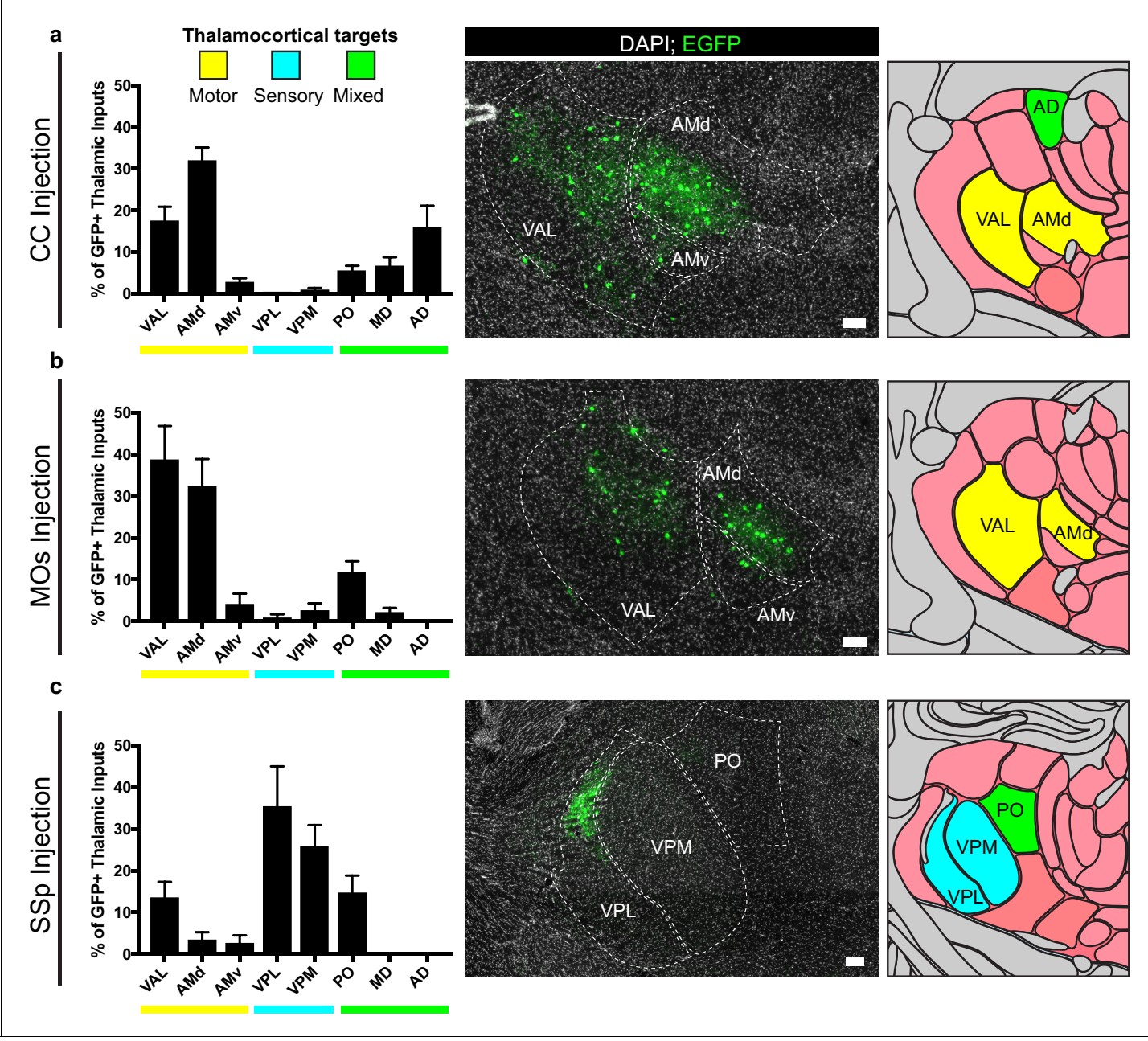

**Figure 5.** Thalamic inputs to OPCs arise from functionally-related nuclei. Tiled immunofluorescence images of GFP+ input neurons were registered to the Allen Brain Atlas to determine the thalamic nuclei from which the inputs arise. Nuclei are color-coded here according to the primary function of their cortical projection targets – motor (yellow), sensory (blue), or mixed (green). (a) Thalamic inputs to OPCs in the CC underlying primary and secondary motor cortices arise primarily from ventral anterior-lateral (VAL) and anteriomedial (AM) nuclei. (b) Thalamic inputs to OPCs in MOs arise primarily from VAL and AM nuclei. (c) Thalamic inputs to SSp arise primarily from ventral posterolateral (VPL) and ventral posteromedial (VPM) nuclei. All bars indicate mean, error bars indicate SEM. Scale bars represent 100 microns. N = 10 mice (CC), eight mice (MOs), and six mice (SSp) respectively. Thalamic nuclei defined as presented in the Allen Brain Atlas and abbreviated as follows: VAL = ventral anterior-lateral, AMd = anteromedial dorsal part, AMv = anteromedial ventral part, VPL = ventral posterolateral, VPM = ventral posteromedial, PO = posterior complex, MD = mediodorsal, AD = anterodorsal.

DOI: https://doi.org/10.7554/eLife.49291.008

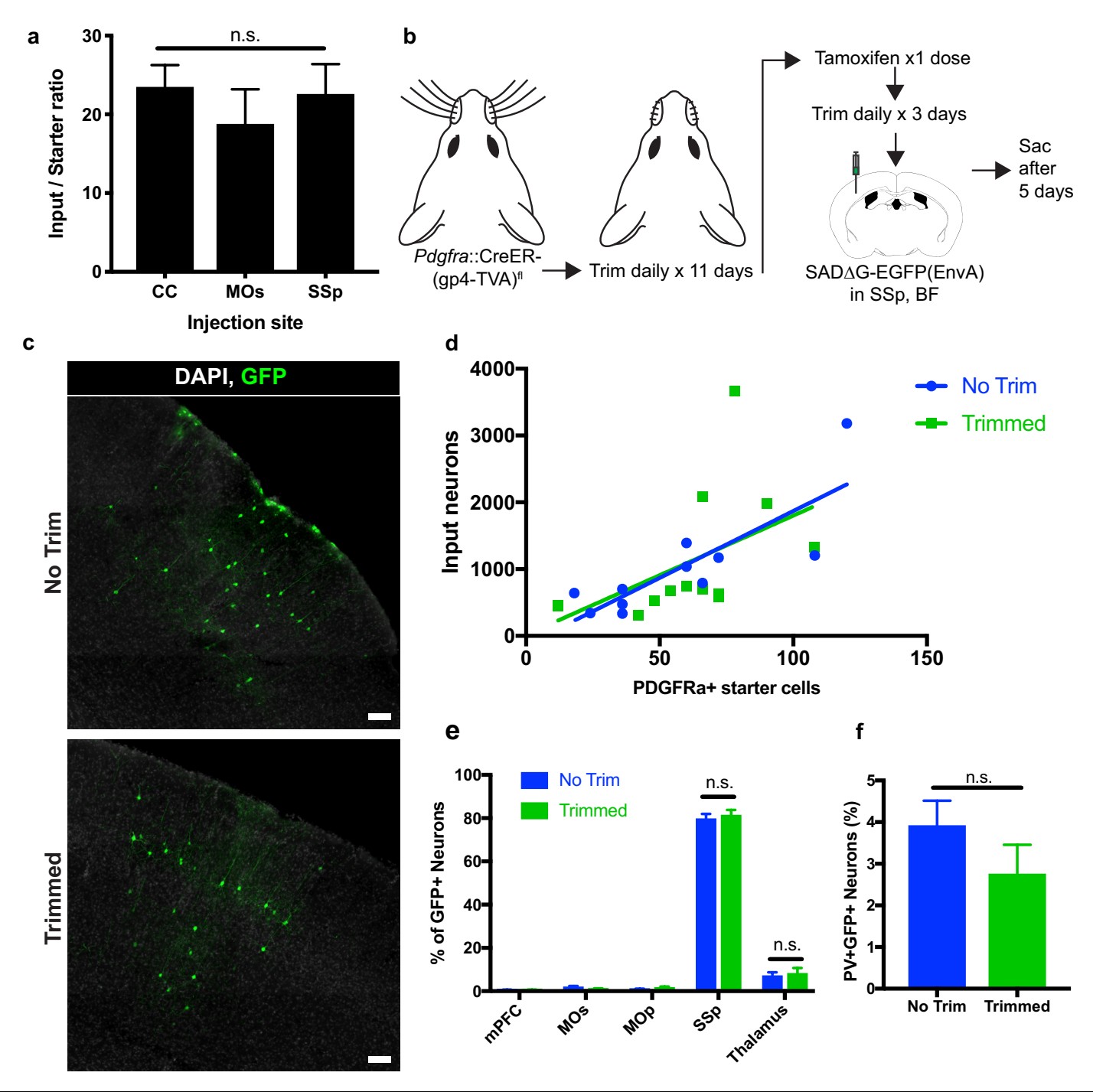

**Figure 6.** Neuronal input to OPC starter ratios are consistent across brain region and despite whisker trimming-induced afferent activity deprivation. (**a**) Neuronal input to OPC starter ratios, as measured by the slope of the best-fit linear regression of GFP+ input neurons against Pdgfra+/GFP+ OPC starter cells. Bars indicate mean, error bars indicate standard error of linear regression. (**b**) Outline of whisker trimming deprivation expriment and subsequent viral injection into barrel field of somatosensory cortex (SSp,BF). (**c**) Representative images of GFP+ input neurons in SSp, BF of non-trimmed and trimmed animals, white = DAPI, green = GFP. (**d**) Scatter plot of GFP+ input neurons against Pdgfra+/GFP+ starter OPCs and best-fit linear regression to assess average input to starter cell ratio in whisker-trimmed (Trimmed) and control (No Trim) groups. Each point represents an independent animal, n = 13 (Trimmed), n = 11 (No Trim). (**e**) Distribution of GFP+ neuronal inputs to OPCs in Trimmed and No Trim groups as a percentage of total inputs. n = 13 (Trimmed), n = 11 (No Trim). (**f**) Proportion of total GFP+ input neurons immunostaining for parvalbumin (PV) in Trimmed and No Trim groups, n = 5 animals each. Bars indicate mean, error bars indicate SEM. Scale bars in (**c**) represent 100 microns. Statistical testing performed by Tukey's multiple comparisons test, n.s. indicates p>0.05.

DOI: https://doi.org/10.7554/eLife.49291.009

While the estimated number of synaptic inputs appears consistent across mapped regions, the localization of these strikingly extensive inputs is distinct by location and supports a pattern of brain-wide afferent connectivity to OPCs from brain regions known to be functionally-associated within the premotor or somatosensory circuits. For OPCs present in the CC genu inferior to the cingulum bundle, there is a relative bias of inputs from cortical regions involved in planning and execution of motor skills, and ~25% of these inputs arise contralateral to the targeted OPCs. While previous studies have demonstrated evoked synaptic inputs to these white matter OPCs by stimulation of callosal fibers, we now provide an unbiased assessment of the cortical projection neurons and interneurons responsible for these synapses, confirm the presence of thalamocortical inputs to OPCs in somatosensory cortex (*Mangin et al., 2012*), and further identify thalamocortical input to OPCs in MOs. Notably, behavioral paradigms shown to alter oligodendrocyte lineage dynamics in mice, including motor learning tasks and social isolation, are thought to drive dynamic changes in neuronal activity in MOs and mPFC. We now demonstrate that the majority of synaptic inputs to callosal white matter OPCs arise from these very brain regions. These synaptic inputs are also largely excitatory, with immunostaining revealing a relatively small fraction of OPC inputs arising from local PV+ interneurons. Strikingly, this fraction is relatively consistent across cortical and white matter territories investigated here, which may result either from higher regional density of excitatory projection axons or may indicate that OPCs actively regulate the number of interneuron inputs as a mechanism to maintain excitatory:inhibitory balance. Relative to transsynaptic tracing studies in neurons, both the relative and absolute degree of inhibitory connectivity is lower in OPCs. Recent studies with transsynaptic tracing from principal neuron starter cells suggest GABAergic inputs range from 18–27% of all inputs in mouse mPFC and 9.5–15% in mouse barrel cortex (*DeNardo et al., 2015*).

Our assessment of synaptic inputs to gray matter OPCs maps neuronal connectivity to this regionally and perhaps functionally distinct cell population. In contrast to callosal white matter OPCs, neuronal inputs to OPCs present in SSp or MOs primarily arise within ipsilateral local cortex, although a smaller fraction (mean approximately 7.5%) of inputs to OPCs in MOs arise contralaterally. Additionally, we show thalamocortical projections from the expected, functionally-associated thalamic nuclei providing synaptic input to these OPCs. Taken together, the demonstrated map of input neurons to cortical OPCs suggests a mechanism by which OPCs could sense synchronized patterns of activity between thalamus and cortex. In turn, integration of this synaptic activity by individual OPCs might coordinate or regulate adaptive myelination of circuitry linking cortical and thalamic territories – a model that merits evaluation in future studies.

While the localization and laterality of neuronal input to OPCs varies depending on brain region, the total numerical extent of input connections – as measured by input:starter ratio – is remarkably consistent across territories. Given that the rate of OPC turnover in these regions has been shown to vary (*Rivers et al., 2008*), it follows that the extent of synaptic input must be regulated by a newly-generated OPC to result in equivalent connectivity. OPCs in afferent activity-deprived somatosensory cortex following unilateral whisker trimming are less likely to survive in a critical temporal window following division, which subsequently results in diminished generation of mature oligodendrocytes (*Hill et al., 2014*). Moreover, genetic ablation of AMPA receptors in OPCs reduces the survival of oligodendrocytes generated during development (*Kougioumtzidou et al., 2017*), and $Ca^{2+}$-permeable AMPARs may have an important role in balancing OPC response to proliferation/differentiation cues (*Chen et al., 2018*). We now demonstrate that deprivation of input activity to barrel field OPCs by whisker trimming does not alter the synaptic input ratios of surviving cells at 15 days after trimming, nor does it impact the distribution of neuronal inputs at the time point evaluated. We selected this timepoint based on 2-photon microscopy studies of spine plasticity in barrel cortex following whisker trimming which have identified new-persistent spine formation in the 12–20 day window after trimming begins (*Knott et al., 2006*; *Holtmaat et al., 2006*). We therefore suspected that a similar window might represent a period of 'new persistent' changes in OPC synaptic connectivity, yet input:starter ratios and input distribution remained similar at the timepoint examined. This may suggest that the pool of OPCs giving rise to early oligodendrocytes in the above studies begin from the same level of synaptic connectivity. From this baseline, activity deprivation-related deficits resulting in decreased survival may accumulate at later stages of cell differentiation. Alternatively, OPCs that fail to attain sufficient synaptic input in the critical window after division may fail to survive, resulting in deficient oligodendrogenesis despite apparently normal starter to

input ratios. Future studies at these early and late timepoints will be required to definitively evaluate potential synaptic remodeling in OPCs following sensory deprivation.

An important limitation of our current study is the inability to delineate the connectivity of single cells, only the population total. This raises the possibility that a mixture of high and low-connectivity OPCs could exist – a possibility supported by the distribution of OPC-colocalized PSD95 puncta we observe in these animals – and newly-generated cells could tend to sort into one pool or the other under the control of local factors. In our hands, we were unable to titrate viral injections to achieve single starter cell transductions. Recently, several groups have achieved transduction of single neurons under 2-photon microscopy guidance using targeted electroporation and/or guided microinjection of rabies virus (*Marshel et al., 2010*; *Wertz et al., 2015*; *Rompani et al., 2017*), suggesting that if technical barriers can be overcome, input networks to single OPCs might be elucidated in the future. This discovery of widespread and remarkably stable neuronal afferents to OPCs thus indicates a need to probe context-specific roles of neuron->OPC synaptic connectivity and ultimately to determine the function of these enigmatic structures.

# Materials and methods

## Key resources table

| Reagent type (species) or resource | Designation | Source or reference | Identifiers | Additional information |
|---|---|---|---|---|
| Antibody | Anti-Pdgfra (goat polyclonal) | R and D systems | AF1062 RRID:AB_2236897 | 1:500 |
| Antibody | Anti-Olig2 (rabbit monoclonal) | Abcam | EPR2673 RRID:AB_10861310 | 1:500 |
| Antibody | Anti-GFP (chicken polyclonal) | Abcam | Ab13970 RRID:AB_300798 | 1:1000 |
| Antibody | Anti-parvalbumin (rabbit polyclonal) | Abcam | Ab11427 RRID:AB_298032 | 1:250 |
| Antibody | Anti-somatostatin (rat monoclonal) | Millipore | MAB354 RRID:AB_2255365 | 1:200 |
| Antibody | Anti-VIP (rabbit polyclonal) | Immunostar | 20077 RRID:AB_572270 | 1:500 |
| Antibody | Anti-Iba1 (rabbit polyclonal) | Wako | 019–19741 RRID:AB_839504 | 1:500 |
| Antibody | Anti-Cre recombinase (mouse monoclonal) | Millipore | MAB3120 RRID:AB_2085748 | 1:1000 |
| Antibody | Anti-PSD95 (rabbit polyclonal) | Invitrogen | 51–6900 RRID:AB_2533914 | 1:100 |
| Antibody | Anti-CASPR (rabbit monoclonal) | Cell Signaling Technologies | 97736 RRID:AB_2800288 | 1:250 |
| Software | Imaris | Bitplane/Oxford Instruments | v8.1.2 RRID:SCR_007370 | |
| Software | Matlab | Mathworks | R2017b RRID:SCR_001622 | |
| Strain (*Mus musculus*) | B6N.Cg-Tg(Pdgfra-cre/ERT)467Dbe/J | The Jackson Laboratory | 018280 RRID:IMSR_JAX:018280 | |
| Strain (*Mus musculus*) | B6;129P2-Gt(ROSA)26Sortm1(CAG-RABVgp4,-TVA)Arenk/J | The Jackson Laboratory | 024708 RRID:IMSR_JAX:024708 | |

## Animal breeding

All animal studies were approved by the Stanford Administrative Panel on Laboratory Animal Care (APLAC). Animals were housed on a 12 hr light cycle according to institutional guidelines. Mice expressing CreER under the control of *Pdgfra* promoter/enhancer regions (*Pdgfra*::Cre/ERT) were

purchased from The Jackson Laboratory (stock number 018280) and have been previously described (*Kang et al., 2010*). Mice expressing a recombinant rabies G glycoprotein gene (*RABV*gp4) along with the gene encoding avian leucosis and sarcoma virus subgroup A receptor (*TVA*) preceded by a *loxP*-flanked STOP fragment and inserted into the *GT(ROSA)26Sor* locus (R26(gp4-TVA)<sup>fl/fl</sup>) have been previously described (*Takatoh et al., 2013*) and were purchased from The Jackson Laboratory (stock number 024708). Hemizygous *Pdgfra*::Cre/ERT mice were then crossed with homozygous R26 (gp4-TVA)<sup>fl/fl</sup> mice to generate animals used in subsequent experiments. Genotyping was performed by PCR according to supplier protocols.

## Viral tracing

EGFP-expressing G-deleted rabies virus pseudotyped with EnvA (SADΔG-EGFP(EnvA)) (*Wickersham et al., 2007*) was prepared at and obtained from the Salk Institute Gene Transfer, Targeting, and Therapeutics Facility vector core (GT3). Virus used in these studies originated in two lots with reported titers of $7.92 \times 10^7$ and $1.94 \times 10^9$ TU/mL. 3 days prior to stereotaxic injections, Cre/ERT-mediated recombination was induced by a single IP injection of 100 mg/kg of tamoxifen (Sigma) solubilized in corn oil. Stereotaxic delivery of virus occurred under isofluorane anesthesia in BSL2+ conditions. 300 nL of SADΔG-EGFP(EnvA) was delivered to the corpus callosum (coordinates AP +1 mm, ML – 1 mm, DV −1.2 mm) or the overlying secondary motor area (coordinates AP + 1 mm, ML – 0.8 mm, DV −0.5 mm) or primary somatosensory cortex (coordinates AP −1 mm, ML −3 mm, DV −0.7 mm) over 5 min (Stoelting stereotaxic injector). Animals were monitored for general health, and no adverse symptoms of viral administration were observed. 5 days following viral injection, animals were deeply anesthetized with tribromoethanol and transcardially perfused with PBS followed by 4% PFA, then brains were removed and post-fixed overnight in 4% PFA. Brains were then transferred to 30% sucrose, and after sinking serial 40 micrometer floating coronal sections were prepared on a freezing-stage microtome for subsequent immunolabeling and imaging.

## Whisker trimming

*Pdgfra*::CreERT; R26(gp4-TVA)<sup>fl</sup> mice generated as described above were trimmed of whiskers bilaterally to the level of the skin using electric clippers daily beginning at P25. At P37, tamoxifen was injected as described above, and whisker trimming continued daily until P40, when SADΔG-EGFP (EnvA) was injected as described above. Animals were then sac'd and perfused at P45 as described above.

## Immunofluorescence and confocal microscopy

Antibodies and dilutions used for immunofluorescence staining were as follows: polyclonal goat anti-mouse Pdgfra (R and D Systems, AF1062, 1:500), monoclonal rabbit anti-mouse Olig2 (Abcam EPR2673, 1:500), polyclonal chicken anti-GFP (Abcam, ab13970, 1:1000), polyclonal rabbit anti-parvalbumin (Abcam, ab11427, 1:250), monoclonal rat anti-somatostatin (Millipore, MAB354, 1:200), polyclonal rabbit anti-VIP (Immunostar 20077, 1:500), polyclonal rabbit anti-Iba1 (Wako, 1:500), mouse anti-Cre recombinase (Millipore, MAB3120, clone 2D8, 1:1000), rabbit anti-PSD95 (Invitrogen, 51–6900, 1:100), and rabbit anti-CASPR (Cell Signaling Technologies, clone D813V). Tissues collected at serial intervals of 1 in every six sections were blocked and permeabilized with 3% normal donkey serum and 0.3% Triton X-100 in Tris-Buffered Saline (3%NDS/TBST) for 30 min at room temperature, followed by incubation with antibodies at the indicated dilution factors in 1%NDS/TBST for 18 hr at four degrees C. For mouse anti-Cre recombinase staining, NDS block was followed by treatment with mouse-on-mouse staining reagent (Vector Laboratories, BMK-2202) prior to incubation with primary antibody. Following a series of washes, secondary AlexaFluor-tagged antibodies raised in donkey (Jackson Immunoresearch) in 1%NDS/TBST were incubated for 4 hr at room temperature, and following a series of washes, sections were counterstained with DAPI (1 ug/mL) and mounted on slides with ProlongGold media (ThermoFisher Scientific). Tile scanning images were acquired at 10X magnification on a Zeiss AxioObserver upright fluorescence microscope with automated stage and tile-scanning capability (Microbrightfield). For identification of atlas regions for labeling quantification, acquired images were manually registered to the closest available section from the Allen Brain Mouse Reference Atlas (*Lein et al., 2007*) (ImageJ) using DAPI fluorescence of the section outline and major neuroanatomical structures to guide fitting. Analysis participants were blinded to injection

conditions, and independent adjustment of atlas registration maps did not substantially impact counting results. Cell counting was performed by two independent reviewers on every 6th 40 micrometer tissue section throughout the brain, and total cell count estimates were derived by multiplying the number of counted cells by 6. Multichannel immunofluorescence microscopy to identify starter cell populations, neuronal identity, and other high-resolution imaging was conducted by acquiring Z-stacks through the target region with a Zeiss LSM710 confocal microscope.

### PSD95 colocalization

Floating sections immunostained for Pdgfra and PSD95 as above were imaged as Z stacks on a Zeiss LSM710 confocal microscope with a 63X objective. Acquired Z-stacks were loaded in Imaris software (Bitplane) with MATLAB integration. Filament models were constructed using the Imaris Filament plugin. PSD95 puncta were selected with the Imaris Spots plugin. Colocalized puncta were identified with the MATLAB plugin for Imaris 'Find spots close to filament.' Modeling parameters were held consistent across individual cohorts.

### Statistics and reproducibility

Stereotaxic injections were repeated in three independent cohorts (litters) of animals for each injection location, and both male and female mice were used. Sample sizes were established based upon similar studies in the literature and were not pre-determined. Cell counters were blinded to injection location, and counts were performed independently by two reviewers. All statistical tests were performed using Graphpad Prism software and details of individual tests are described in figure legends.

## Acknowledgements

We thank Brady Weissbourd for helpful discussion and Pamelyn Woo for assistance with animal colony maintenance. The authors gratefully acknowledge support from the California Institute for Regenerative Medicine (CIRM RN3-06510), National Institute of Neurological Disorders and Stroke (NINDS R01NS092597 and F31NS098554), NIH Director's Pioneer Award (DP1NS111132), SFARI Foundation, Maternal and Child Health Research Institute at Stanford.

## Additional information

### Funding

| Funder | Grant reference number | Author |
| --- | --- | --- |
| NIH Office of the Director | DP1NS111132 | Michelle Monje |
| California Institute for Regenerative Medicine | CIRM RN3-06510 | Michelle Monje |
| National Institute of Neurological Disorders and Stroke | R01NS092597 | Michelle Monje |
| National Institute of Neurological Disorders and Stroke | F31NS098554 | Christopher W Mount |
| Simons Foundation | The Simons Foundation Autism Research Initiative | Belgin Yalçın Michelle Monje |
| Stanford University | Maternal and Child Health Research Institute | Michelle Monje Belgin Yalçın |

The funders had no role in study design, data collection and interpretation, or the decision to submit the work for publication.

### Author contributions

Christopher W Mount, Conceptualization, Formal analysis, Investigation, Writing—original draft; Belgin Yalçın, Kennedy Cunliffe-Koehler, Formal analysis, Investigation, Writing—review and editing;

Shree Sundaresh, Software, Investigation, Methodology; Michelle Monje, Conceptualization, Supervision, Funding acquisition, Writing—original draft

**Author ORCIDs**
Christopher W Mount (ID) https://orcid.org/0000-0003-1635-2492
Belgin Yalçın (ID) https://orcid.org/0000-0001-9689-9062
Michelle Monje (ID) https://orcid.org/0000-0002-3547-237X

**Ethics**
Animal experimentation: This study was performed in strict accordance with the recommendations in the Guide for the Care and Use of Laboratory Animals of the National Institutes of Health. All animal studies were performed in accordance with a protocol approved by the Stanford Administrative Panel on Laboratory Animal Care (APLAC, protocol number 27215).

**Decision letter and Author response**
Decision letter https://doi.org/10.7554/eLife.49291.013
Author response https://doi.org/10.7554/eLife.49291.014

## Additional files

**Supplementary files**
• Source data 1. Raw data presented in the paper.
DOI: https://doi.org/10.7554/eLife.49291.010
• Transparent reporting form DOI: https://doi.org/10.7554/eLife.49291.011

**Data availability**
The data that support the findings of this study are presented in Source data 1.

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
