## [Decision Letter]

Thank you for submitting your article "Monosynaptic tracing maps brain-wide afferent oligodendrocyte precursor cell connectivity" for consideration by *eLife*. Your article has been reviewed by three peer reviewers, and the evaluation has been overseen by a Reviewing Editor and Gary Westbrook as the Senior Editor. The following individual involved in review of your submission has agreed to reveal their identity: David A Lyons (Reviewer #1). The reviewers have discussed the reviews with one another and the Reviewing Editor has drafted this decision to help you prepare a revised submission.

Summary:

Using a modified rabies virus to specifically label "presynaptic" neurons by an unbiased retrograde monosynaptic tracing approach, Mount et al. map out the neuronal inputs that oligodendrocyte precursor/ progenitor cells (OPCs) receive in three different anatomical locations in the forebrain of mice (corpus callosum, motor cortex and somatosensory cortex). Although it has been known for quite some time that OPCs receive synaptic input from both excitatory and inhibitory neurons, the techniques employed in this manuscript provide the opportunity to further explore this circuitry. The authors provide a robust dataset confirming the connectivity between OPCs in white matter and grey matter regions with various neurons and neuronal subtypes across the brain.

The greatest strength of this study is the strategy employed for monosynaptic tracing of connectivity. On the other hand, the study still misses the individual variance and in-depth information on whether certain types of OPCs receive mainly glutamatergic or GABAergic inputs, or whether GABAergic inputs to OPCs are more frequent in certain layers of the cortex. This information would be helpful as some oligodendrocytes myelinate only GABAergic, or glutamatergic neurons, whereas others myelinate both. Given the large number of labelled OPCs it is also surprising that no neuromodulatory input, such as cholinergic synapses, were detected. Overall, the findings complement previously published works. The novelty is in the methodology, not the insight of the input that OPCs receive.

Overall, the reviewers thought that the conclusion (including the negative data), see below) were a little bit overstated and should be toned down. Terms, such as "functionally related" and "functionally associated" suggest a specificity of OPC targeting that remains unproven, as it may merely reflect the specificity of underlying neuronal circuitry, with synaptic targeting of the omnipresent OPC being a secondary effect (see also question 2 below).

Essential revisions:

1) What proportion of OPCs have synapses? How does the overall number of neuron-OPC connections relate to the structure of individual OPCs? Antibody-based staining of postsynaptic components and imaging of OPCs at highest resolution should reveal what proportion of OPCs make connections with axon-like profiles? Similarly, and perhaps a bit easier, it should be possible to count post-synaptic clusters per OPC and relate that to the overall ratio predicted from the connectivity maps. The data in general related to the ratio of inputs to OPCs look very convincing, but the left-most value in Figure 2C is unclear. It looks like very few OPCs contact almost 5000 neurons. Can the authors clarify that point?

2) Are the neurons that make synapses with OPCs strictly unmyelinated or do they include myelinated or partially myelinated cells? That issue could also be addressed by immunostaining of labelled neurons.

3) How easy is it to control the mosaicism, i.e. the number/density of the starting population? It would be useful if the authors provide more details on the method throughout, which could make this manuscript potentially suitable for *eLife*'s Tools and Resources format. A more detailed schematic of the molecular mechanism behind the tracing strategy would also be helpful).

4) One experimental concern was the possibility of off-target recombination of neurons using the PDGFRa-CreERT2 line. Although the authors note that Cre protein expression was only detectable in OPCs, sporadic recombination in neurons has been reported in the PDGFRa-CreERT2 lines from the Richardson lab line as well as the Bergles lab (used in this study). It seems unlikely that off-target neuronal recombination and subsequent rabies virus infection would occur in a widespread manner, but it is important to categorically rule out this possibility, given that the entire paper hinges on the neuronal populations expressing GFP being infected trans-synaptically rather than directly. Staining for Cre protein is an imperfect approach as sub-detectable amounts of protein may still be sufficient to drive recombination. Can staining or ISH for the gp4 protein be used to confirm that GFP-expressing neurons were not recombined and infected directly? Alternatively, a very early time-point could be assessed post-infection to confirm GFP expression in the OPC starter cells but not (yet) in any presynaptic neuron.

5) One point that the authors were unable to address is the connectivity patterns of individual OPCs. The authors' comment that it "cannot be tested with existing methods" – but is that really the case? Can the authors elaborate in the Discussion about how one might go about tackling this issue, e.g. generate animals with very few starting OPCs, applying methods to automate tracing across what are very large datasets (not suggesting that the authors need to trace these inputs as part of the current paper).

6) How does the excitatory-inhibitory ratio of input onto OPCs compare with actual excitatory-inhibitory ratios of neurons? Can the authors comment on whether their observed ratios represent a skew towards specific neuronal subtypes? It is also interesting that there appears to be input from neurons that are not myelinated. Perhaps the authors could speculate on possible functions associated with such connectivity.

7) The manuscript would benefit from further information on where exactly the transfected OPCs are located. For example, was the highest proportion of infected OPCs located in layer 4? This information is essential to interpret the data and understand the anatomy of the neuronal inputs to OPCs.

8) The author states "These previously unidentified thalamic inputs are…" Then the author states "These previously unidentified thalamocortical synaptic inputs arise from functionally-related thalamic nuclei." And again the author states "…as well as previously unrecognized OPC inputs from thalamocortical projections." These statements are surprising given that the thalamocortical inputs in OPCs have been well described by Gallo's group. Particularly, in focusing on the barrel cortex, they have shown functional thalamic cortical inputs to OPCs, by stimulating the VB area of the thalamus and record the inputs in the OPCs at different locations within the cortex, and they found a difference in the number of thalamic inputs OPCs receive depending on their location within the cortex (Mangin et al., 2012). These statements need to be rephrased to adhere to the literature.

9) The whisker trimming experiment was carried out from one time-point with analysis at one other time. Is this a period over which other aspects of cellular and functional remodelling occur in the barrel field? Reviewers were unsure how much to take from the negative data shown. Moreover, the lack of an effect is difficult to interpret when observing a large population of cells in a large cortical area. Another concern is the duration of the repeated trimming, as in adult mice, some plasticity changes in the barrel cortex take more time to be detected than those in younger mice (in some papers testing for synaptic structural changes in adult mice, which are often found to layer specific, use repeated trimming protocol for up to 4 weeks). Thus, it needs to be discussed whether the method is sensitive enough to detect regional and small changes in synaptic inputs.

---

## [Author Response]

Summary:Using a modified rabies virus to specifically label "presynaptic" neurons by an unbiased retrograde monosynaptic tracing approach, Mount et al. map out the neuronal inputs that oligodendrocyte precursor/ progenitor cells (OPCs) receive in three different anatomical locations in the forebrain of mice (corpus callosum, motor cortex and somatosensory cortex). […]Overall, the reviewers thought that the conclusion (including the negative data), see below) were a little bit overstated and should be toned down. Terms, such as "functionally related" and "functionally associated" suggest a specificity of OPC targeting that remains unproven, as it may merely reflect the specificity of underlying neuronal circuitry, with synaptic targeting of the omnipresent OPC being a secondary effect (see also question 2 below).

We are grateful for the thoughtful comments. We have now made significant updates based on these helpful suggestions: We have further confirmed specificity of labeling by adding control non-EnvA-bearing tracer injections (Figure 1—figure supplement 1). We performed PSD95 puncta colocalization studies as a quantitative estimate of potential synaptic densities on OPCs, and find that this estimate correlates well with the input:starter cell ratios that we measured throughout our transsynaptic tracing experiments (Figure 2—figure supplement 1). We have also performed additional studies and made textual modifications as suggested, including clarifying that our intended meaning of “functionally-related/associated” was simply that we observed inputs to OPCs from the expected regions of brain known to be functionally related, not to suggest a specificity of OPC targeting apart from this. Specific responses to reviewer comments are detailed below:

Essential revisions:1) What proportion of OPCs have synapses? How does the overall number of neuron-OPC connections relate to the structure of individual OPCs? Antibody-based staining of postsynaptic components and imaging of OPCs at highest resolution should reveal what proportion of OPCs make connections with axon-like profiles? Similarly, and perhaps a bit easier, it should be possible to count post-synaptic clusters per OPC and relate that to the overall ratio predicted from the connectivity maps.

With regard to the synapse density per OPC question, we sought to address this as suggested by immunostaining for PSD95 and assessing for co-localization with OPC processes as defined by PDGFRa expression (subsection “OPCs in corpus callosum receive brain-wide synaptic input”). On average, approximately 20 PSD95+ puncta co-localize with 3D models of OPC processes generated from PDGFRa+ immunostaining; this density is not significantly different from the average input:starter cell ratio calculated by linear regression from the transsynaptic tracing data (Figure 2—figure supplement 1). On the level of individual cells, the distribution of PSD95+ colocalized puncta ranges from 0-67 across the 95 individual cells imaged; ~95% of OPCs analyzed in the corpus callosum exhibit PSD95+ puncta. While PSD95 would not account for potential inhibitory inputs, overall the similarity between the two approaches further supports our earlier estimates. We thank the reviewer for this helpful question.

The data in general related to the ratio of inputs to OPCs look very convincing, but the left-most value in Figure 2C is unclear. It looks like very few OPCs contact almost 5000 neurons. Can the authors clarify that point?

Thank you for pointing out this important point to address. The above PSD95 puncta analysis also suggests that with regard to Figure 2C, there are not individual OPCs making 5000 connections. Rather, we suspect that regional effects in the corpus callosum (CC) – either increased relative OPC cell density, enhanced susceptibility of CC OPCs to viral-induced toxicity, or increased relative rate of OPC differentiation in the CC contributed to systematic underestimates of the true starter cell ratio, resulting in an overall left-shift of the input:starter ratio curve and hence a y-intercept > 0. To address this, we performed diluted rabies virus injections at a 1:5 ratio to better sample low starter cell ratios and attempt to reduce regional effects (Figure 2C). This approach was somewhat successful, and the y-intercept of the revised curve is now 1610 without a substantial impact on the calculated starter:input ratio. Therefore, it is likely that we are still missing starter cells, and further dilutions of virus did not lead to reliable labeling patterns useable for quantification. We have now discussed this limitation in the Discussion section.

2) Are the neurons that make synapses with OPCs strictly unmyelinated or do they include myelinated or partially myelinated cells? That issue could also be addressed by immunostaining of labelled neurons.

This is an excellent question. We observe transsynaptic labeling of both myelinated and unmyelinated input neurons (subsection “OPCs in corpus callosum receive brain-wide synaptic input”). To better illustrate this, we performed immunofluorescence labeling with CASPR to identify paranode/juxtaparanode of myelinated axons (Figure 1—figure supplement 1E). Many labeled axons are devoid of any CASPR labeling through their visualized path, consistent with previous reports of synapses with unmyelinated callosal axons.

3) How easy is it to control the mosaicism, i.e. the number/density of the starting population? It would be useful if the authors provide more details on the method throughout, which could make this manuscript potentially suitable for eLife's Tools and Resources format. A more detailed schematic of the molecular mechanism behind the tracing strategy would also be helpful).

Mosaicism of the starting population can be controlled to some extent by adjusting the amount of total viral particles delivered, as we describe above. Optimizing for specific starter cell densities also depends on the native cell density at the target site, as a sufficient volume must be infused to propagate virus through the target site. We have updated the text to offer more description of the methodology as suggested (subsection “Development and validation of retrograde monosynaptic OPC tracing strategy”). We have also updated the diagram in Figure 1A to provide additional clarification as suggested.

4) One experimental concern was the possibility of off-target recombination of neurons using the PDGFRa-CreERT2 line. Although the authors note that Cre protein expression was only detectable in OPCs, sporadic recombination in neurons has been reported in the PDGFRa-CreERT2 lines from the Richardson lab line as well as the Bergles lab (used in this study). It seems unlikely that off-target neuronal recombination and subsequent rabies virus infection would occur in a widespread manner, but it is important to categorically rule out this possibility, given that the entire paper hinges on the neuronal populations expressing GFP being infected trans-synaptically rather than directly. Staining for Cre protein is an imperfect approach as sub-detectable amounts of protein may still be sufficient to drive recombination. Can staining or ISH for the gp4 protein be used to confirm that GFP-expressing neurons were not recombined and infected directly? Alternatively, a very early time-point could be assessed post-infection to confirm GFP expression in the OPC starter cells but not (yet) in any presynaptic neuron.

Due to the limited ability to directly stain for TVA or gp4 (we inquired after aliquots of previously-described anti-TVA antibodies, which are not available commercially and seem to have fallen into disuse in host laboratories, and available gp4 antibodies are non-reactive in our tissue) we sought to address this by comparison injection of non-EnvA SADΔG19. In the absence of EnvA, viral uptake is no longer restricted to TVA-expressing OPCs and can be taken up more broadly. As now illustrated in Figure 1—figure supplement 1D, with injection of SADΔG19 into primary somatosensory cortex we observe broad uptake throughout the entirety of the injection site in both Cre+ and Cre negative animals with little discernable difference between the two (subsection “Development and validation of retrograde monosynaptic OPC tracing strategy”). By contrast, SADΔG19-EnvA exhibits minimal labeling in Cre WT animals (Figure 1B) – which we would now expect if there were substantial leak of TVA expression in neurons.

5) One point that the authors were unable to address is the connectivity patterns of individual OPCs. The authors' comment that it "cannot be tested with existing methods" – but is that really the case? Can the authors elaborate in the Discussion about how one might go about tackling this issue, e.g. generate animals with very few starting OPCs, applying methods to automate tracing across what are very large datasets (not suggesting that the authors need to trace these inputs as part of the current paper).

While this remains beyond our current capabilities in the specific context of OPCs, single cell network tracing has been achieved by several groups in neurons and some discussion reveals the challenges that must be overcome before this could be approached in OPCs. Both (Marshel et al., 2010) and (Wertz et al., 2015) achieve single-cell access by targeted electroporation of plasmids into single neurons by “shadow patching” under two-photon (2P) guidance in mice (Kitamura et al., 2008, Nat Meth). While to our knowledge this has not been attempted in OPCs, several obstacles are apparent. First, OPCs in our hands at least have been challenging cells to electroporate after isolation forin vitro studies, and substantial optimization may be necessary to achieve efficiencies compatible with the multi-plasmid electroporation necessary to both label the starter cell and provide the requisite packaging and payload elements for transsynaptic tracing. Additionally, the ability of OPCs to divide and dilute any epigenomic plasmids may limit one’s ability to subsequently track the starter cell. More recently, (Rompani et al., 2017) achieved single-cell-initiated tracing in the LGN by a combination of targeted electroporation and guided microinjection of rabies virus under 2P guidance targeting individual principal cells. In summary, 2P guidance might be utilized in the future to guide single-OPC targeted transduction, potentially with the assistance of automated tracing methods to assess distant inputs. The reviewer’s point is well-taken and we have updated the Discussion section accordingly to reflect this (Discussion, last paragraph).

6) How does the excitatory-inhibitory ratio of input onto OPCs compare with actual excitatory-inhibitory ratios of neurons? Can the authors comment on whether their observed ratios represent a skew towards specific neuronal subtypes? It is also interesting that there appears to be input from neurons that are not myelinated. Perhaps the authors could speculate on possible functions associated with such connectivity.

On a structural level, excitatory/inhibitory input ratios to neurons have been measured in several recent trans-synaptic tracing studies. In their study of mouse mPFC and barrel cortex, DeNardo et al. report ratios ranging from 18-27% GABAergic inputs to principal neurons across cortical layers of mPFC, and between 9.5-15% in mouse barrel cortex (DeNardo et al., 2015). In a more recent study by Yetman et al., in SS cortex supragranular principal neurons, a reported average of 7.4 +/- 1.2 parvalbumin inputs to individual principal neurons are present (Yetman et al., Nat Neuro, 2019). Compared with these studies, both the absolute number and fractional percentage of inhibitory inputs to OPCs are lower than seen in neurons in these territories, and we have updated the text to reflect this (Discussion).

7) The manuscript would benefit from further information on where exactly the transfected OPCs are located. For example, was the highest proportion of infected OPCs located in layer 4? This information is essential to interpret the data and understand the anatomy of the neuronal inputs to OPCs.

We thank the reviewers for this suggestion, and have consequently included measurements of cortical location for starter cells in CC, MOs, and SSp (see subsection “OPCs in corpus callosum receive brain-wide synaptic input”; Figure 1—figure supplement 1F). These data demonstrate that cortical distribution of targeted starter cells in both MOs and SSp are broad without evidence of layer-specific targeting. With our injection strategy, this was a necessary trade-off to balance adequate starter cell transduction against injection site specificity. OPCs within layer 4 are indeed sampled, but do not comprise a dominant fraction and we feel there is insufficient resolution at this time to comment more thoroughly on OPC inputs at the level of cortical microcircuitry.

8) The author states "These previously unidentified thalamic inputs are…" Then the author states "These previously unidentified thalamocortical synaptic inputs arise from functionally-related thalamic nuclei." And again the author states "…as well as previously unrecognized OPC inputs from thalamocortical projections." These statements are surprising given that the thalamocortical inputs in OPCs have been well described by Gallo's group. Particularly, in focusing on the barrel cortex, they have shown functional thalamic cortical inputs to OPCs, by stimulating the VB area of the thalamus and record the inputs in the OPCs at different locations within the cortex, and they found a difference in the number of thalamic inputs OPCs receive depending on their location within the cortex (Mangin et al., 2012). These statements need to be rephrased to adhere to the literature.

We thank the reviewers for their attention to this unintended implication, as it was our desire to highlight these findings in secondary motor area and the language was inadvertently carried forward in subsequent discussion of thalamocortical projections to barrel cortex. We have rephrased these statements appropriately and discuss the Mangin et al., 2012 paper prominently (Discussion, second paragraph).

9) The whisker trimming experiment was carried out from one time-point with analysis at one other time. Is this a period over which other aspects of cellular and functional remodelling occur in the barrel field? Reviewers were unsure how much to take from the negative data shown. Moreover, the lack of an effect is difficult to interpret when observing a large population of cells in a large cortical area. Another concern is the duration of the repeated trimming, as in adult mice, some plasticity changes in the barrel cortex take more time to be detected than those in younger mice (in some papers testing for synaptic structural changes in adult mice, which are often found to layer specific, use repeated trimming protocol for up to 4 weeks). Thus, it needs to be discussed whether the method is sensitive enough to detect regional and small changes in synaptic inputs.

We agree with the reviewers that definitive evaluation of potential synaptic remodeling will require systematic analysis at multiple additional timepoints. For the purposes of our study, we sought to select a timepoint that was both amenable to input:starter ratio quantification and consistent with prior studies of persistent synaptic alterations – at least on a structural level – from prior studies in mouse barrel cortex. Regarding animal age, elegant work (Hill et al., 2014) has shown that whisker trimming in the early postnatal (P6-8) window over a short timecourse of several days reduced survival of differentiating OPCs, suggesting that studies of input:starter cell ratios would be complicated in this window – an already challenging one given that there is substantial oligodendrocyte lineage expansion and differentiation underway. Therefore, we decided to perform our study in young adult animals. As far as duration of whisker trimming, (Hughes et al., 2018) demonstrated that sensory enrichment for 21 days in adult mice enhanced oligodendrocyte generation that was abrogated with whisker trimming, but found no effect with whisker trimming alone. Thus, without definite guidance from the OPC literature, we turned to the synaptic plasticity literature with special attention to structural studies of spine generation and persistence in barrel cortex following whisker trimming. The seminal work of (Holtmaat et al., 2006) assessed this question with whisker trimming in mice with a median age of 2.3 months and evaluated spine formation in barrel cortex pyramidal neurons by 2P microscopy daily after the initiation of trimming. They binned newly-formed spines as those appearing within 0-12 days of trimming initiation, and new-persistent spines as those continuing to be present through 12-20 days of trimming. Additional studies, including (Knott et al., 2006) further suggest that persistent spines – in particular those lasting 4 days or longer, are more likely to acquire synapses. On this basis, we felt that a total trimming period of 14 days was reasonable to assess a possible period of “new persistent” changes in OPC synaptic connectivity. However, we are of course unable to rule out alterations to connectivity in other temporal domains or brain regions, which could be a valuable area of investigation. We have updated the text to reflect this discussion (Discussion, fourth paragraph). We have also clarified throughout that our data represent only one timepoint.